# A Thermopile-Based Gas Flow Sensor with High Sensitivity for Noninvasive Respiration Monitoring

**DOI:** 10.3390/mi14050910

**Published:** 2023-04-23

**Authors:** Zemin Liu, Chenchen Zhang, Xuefeng Ding, Yue Ni, Na Zhou, Yanhong Wang, Haiyang Mao

**Affiliations:** 1School of Instrument and Electronics, North University of China, Taiyuan 030051, China; 2Institute of Microelectronics of Chinese Academy of Sciences, Beijing 100029, China; 3Jiangsu Hinovaic Technologies Company Ltd., Wuxi 214135, China

**Keywords:** gas flow sensor, microheater, N/P polySi thermopile, respiration monitoring

## Abstract

In this work, a N/P polySi thermopile-based gas flow device is presented, in which a microheater distributed in a comb-shaped structure is embedded around hot junctions of thermocouples. The unique design of the thermopile and the microheater effectively enhances performance of the gas flow sensor leading to a high sensitivity (around 6.6 μV/(sccm)/mW, without amplification), fast response (around 35 ms), high accuracy (around 0.95%), and mood long-term stability. In addition, the sensor has the advantages of easy production and compact size. With such characteristics, the sensor is further used in real-time respiration monitoring. It allows detailed and convenient collection of respiration rhythm waveform with sufficient resolution. Information such as respiration periods and amplitudes can be further extracted to predict and alert of potential apnea and other abnormal status. It is expected that such a novel sensor could provide a new approach for respiration monitoring related noninvasive healthcare systems in the future.

## 1. Introduction

In recent years, disability and morbidity caused by respiration diseases, which are second only to cardiovascular diseases, impose immense economic costs and health burden all over the world [1]. Respiration is an essential process for gas exchange between the internal and external environment of a human body, which plays an important role in maintaining normal physiological functions. Respiration states (including breathing frequency, depth, time ratio of each stage in a single breath, etc.), are usually utilized to reflect the health condition of a human body [2].

Reliable and accurate respiration monitoring can effectively predict and diagnose potential diseases such as apnea, asthma, and chronic obstructive pulmonary [3,4,5]. A gas flow sensor can directly sense the airflow during respiration [6,7] and thus is suitable for monitoring the respiration states. There are several types of flow sensors, such as turbine flow sensors [8], fiber optic-based flow sensors [9,10], piezoresistive/piezoelectric flow sensors [11,12], and thermal flow sensors [13]. The sensors of the first two types need to be matched with complex filter, rectifier, and other pretreatment devices when used in respiration monitoring, which is not conducive to the miniaturization of the system. On the other hand, the latter two types can be fabricated by adopting micromachining technology [14], which thus can be largely scaled down. For a piezoresistive or piezoelectric flow sensor, the deformation of cantilevers induced by the breathing airflow can reflect the respiration states, whereas the moving parts in these sensors are susceptible to irreversible damage when they receive excess momentum from the airflow [15]. In addition, they have difficulty in mass production due to their complex processing. As is different from the piezoresistive/piezoelectric flow sensors, a thermal flow sensor usually consisting of a heating structure and a temperature-sensitive structure, measuring the flow velocity through the heat exchange of the airflow, and as such a sensor is easier to fabricate, and it has better stability as it involves no movable structures.

Conventionally, there are two types of thermal flow sensors, namely the thermistor flow sensors [16,17,18] and the thermopile flow sensors [19,20]. For the thermistor-based flow sensors, their fabrication process is relatively simple and low-cost, but complicated external circuits are usually required to convert resistance variation to voltage signals. Compared with the thermistor-based flow sensors, the Seebeck effect-based thermopile flow sensors can be fabricated through a complementary metal oxide semiconductor (CMOS) compatible process as well, but they require no complex peripheral circuits [21] and thus are superior when used for airflow sensing. For a thermopile-based gas flow sensor, improving the thermoelectric conversion efficiency of the thermopile and enhancing the heat exchange between the airflow and the heat source are two important routes for the performance optimization. To date, most of the previously reported thermopile-based gas flow sensors have used p + Si/metal thermocouples [22,23]. However, the large thermal conductivity of metal tends to reduce the temperature difference between the hot and cold junctions of the thermopile, which is not favorable to thermoelectric conversion and stability. In addition, in order to increase the heat exchange efficiency between the heat source and the airflow, the design of the microheater structure has undergone different forms from linear [24] and annular [25] to serpentine shapes [26]. However, the contact surfaces between the high temperature region and the airflow formed by these microheaters have still been relatively small, which restrained the heat exchange efficiency.

In this work, a stacked N/P polySi thermopile and a comb-shaped microheater are adopted to form a gas flow sensor, where the hot junctions of the thermopile are wrapped between the comb teeth of the microheater. The stacked N/P polySi thermopile has a high thermoelectric conversion efficiency and thus is more sensitive to temperature changes caused by airflow. Meanwhile, the comb-shaped microheater generates a larger area of high temperature to enhance the heat exchange efficiency with the airflow. By synergistically enhancing the thermoelectric conversion efficiency of the thermopile and the heat exchange efficiency between the microheater and the measured airflow, the performance of the designed gas flow sensor is significantly improved. With these efforts, the sensor achieves a high sensitivity of 414 μV/sccm (6.6 μV/sccm/mW), a fast response time of 35 ms, and a high accuracy of 0.95%. In addition, the sensor is able to respond to human breathing, and respiration states of a human body can be effectively obtained according to the waveform of the output signals of the sensor. It indicates great potential of the sensor in the applications of noninvasive respiration monitoring.

## 2. Materials and Methods

### 2.1. Design and Fabrication

Figure 1 shows the schematic diagram of the structure and principle of the thermopile-based gas flow sensor. Around the hot junctions of the thermopile, there is distributed a comb-shaped microheater, which is used to provide Joule heat thus to generate a gradient of temperature field between the hot junctions and the cold junctions of the thermopile. In this case, the air domain around the microheater forms a stable thermal field above the ambient temperature through natural convection heat transfer when there is no airflow. As the airflow increases, forced convection heat transfer gradually dominates, and the heat in the high temperature domain is absorbed and transported by the airflow, resulting in a decrease in temperature near the hot junctions of the thermopile. As the flow rate increases, the temperature gradient on the sensor chip surface decreases, and thus the sensor output decreases accordingly. Thus, the basic principle of the thermopile-based gas flow sensor responding to the airflow can be expressed by the following formula: (1)$$\Delta T=T_{\mathrm{diff}1}-T_{\mathrm{diff}2}$$
(2)$$U_{static}=(\alpha_{P}-\alpha_{N})T_{\mathrm{diff}1}$$
(3)$$U_{flow}=(\alpha_{P}-\alpha_{N})T_{\mathrm{diff}2}$$
where $T_{\mathrm{diff}1}$, $T_{\mathrm{diff}2}$ are the temperature difference between the cold junctions and the hot junctions of the thermopile without and with airflow, respectively. $U_{\mathrm{static}}$ is the sensor output voltage without airflow (static); $U_{\mathrm{flow}}$ is the sensor output voltage at a certain flow rate; $\alpha_{N}$, $\alpha_{P}$ correspond to the Seebeck coefficient of the N and P type polySi, respectively.

The fabrication process of the device is shown in Figure 2, which is compatible with CMOS process. Firstly, SiO_2_-Si_3_N_4_-SiO_2_ multilayers as a supporter with stress control were successively deposited on a silicon wafer by using low-pressure chemical vapor deposition (Figure 2a). Secondly, polySi layers were deposited, doped, and patterned to form N type and P type polySi strips, respectively. Detailed information about the steps of deposition, injection, thermal annealing, and dry etching was reported in our previous work [27]. The N/P polySi strips were distributed in a stacked structure to form a series of thermocouples, which effectively utilizes the sensitive area (Figure 2b,c). Then, an etching step was performed to form contact vias for electrical connection (Figure 2d). Next, an Al layer was sputtered and patterned to form the microheater, the electrodes, and the electrical connection lines at the same time, followed by deposition of SiO_2_ as an isolation layer (Figure 2e,f). After that, a Si_3_N_4_ layer was deposited on the top as passivation for critical structures, and the low thermal conductivity of Si_3_N_4_ also facilitated the flow to absorb heat from the microheater (Figure 2g). Finally, the sensor was released from the backside by deep reactive ion etching as shown in Figure 2h. The main structural parameters of the sensor are shown in Table 1.

Scanning electron microscope (SEM) images of the fabricated flow sensor are shown in Figure 3. From the top view shown in Figure 3a, the overall dimensions of the sensor are 1650 μm × 1650 μm. Figure 3b clearly displays the cross-section view of a stacked N/P polySi thermocouple. As can be observed in Figure 3c,d, the microheater is embedded around the hot junctions of the thermocouples in a comb-shaped structure, and in the thermopile, the P polySi strips are slightly narrower than the N polySi strips. Ascribe to the high uniformity of the process, the relative standard deviation of the resistance of N/P polySi thermopile can be controlled within 1.1% according to the wafer-level chip probing test. This result is obtained by statistical analysis of N/P polySi thermopile resistance values (around 2217 devices) for the same structure on the wafer. It confirms that the sensors prepared by the aforementioned process have good consistency and are suitable for mass production.

### 2.2. Experimental Setup

The measurement setup is depicted in Figure 4. The instruments used in the experiments include a voltage source (Keithley, 2450 SMU), a digital multimeter (Tektronix, DMM 6500 6^1/2^), and an air–liquid distribution system (ELITE TECH, DGL-Ⅲ). A constant voltage source was used to provide a stable heating voltage, and the digital multimeter read and displayed the sensor’s voltage. The airflow source was provided by a commercial high-pressure cylinder pf pure liquid nitrogen(N_2_) with a capacity of 40 L. The N_2_ flow was quantitatively regulated by means of the laboratory gas distribution control system and related to upper computer software. According to the formula of the Reynolds number (Re) as follows
(4)Re=ρvd/η
where ρ is the gas density; η is the gas viscosity coefficient; v is the airflow velocity; and d is the characteristic length of the flow channel. Accordingly, the maximum Reynolds number was calculated to be 1058.1 when N_2_ was chosen as the experimental gas. Since the Reynolds number was smaller than 2300, the flow state in the pipeline could be regarded as laminar flow [28]. Therefore, the distribution of gas velocity in the gas channel could be considered as being regular, and the velocity of the gas gradually increased from the edge to the center. In order to obtain the optimized test results, the sensor was placed in the center of the pipe during the experiment.

## 3. Results and Discussion

### 3.1. Basic Performance Characterization of the Thermopile-Based Gas Flow Sensor

The actual measurement system is shown in Figure 5. In the system, the sensor was wire bonded on top surface of a transistor outline (TO) package header, and a cuboid flow pipe with an inner area of 10 mm × 10 mm was designed and fabricated using an acrylic sheet. The output characteristics of the flow sensor are shown in Figure 6, where Figure 6a illustrates the output of the thermopile device under different heating voltages in the absence of flow, indicating that the sensor can accurately respond to different input voltages. When the microheater is worked at a constant voltage of 5 V, it forms a temperature gradient of about 200 °C between the center and the edges of the temperature field. As is shown in Figure 6b, the output of the device decreases with increasing of the gas flow rate. This is because when the flow rate increases, more heat around the hot junctions is carried away by the gas; thus, the temperature difference between the hot junctions and cold junctions is decreased and finally leads to a decrease in the output voltage of the thermopile device. As can be seen from the curve, the output voltage changes linearly with the flow rate at a range of 200–900 sccm. In the low and high flow velocity regions, little nonlinearity can be observed in the response curves, due to the way of convective heat transfer and heat exchange capacity. Accordingly, a normalized sensitivity of 6.6 μV/sccm/mW can be obtained which is much higher than that of the previously reported thermopile-based gas flow sensors, as show in Table 2. This is assumed to be benefited from the stacked heavily doped N/P polySi thermocouples in such a design; the number of thermocouples in a limited area is doubled; thus, the sensitivity to temperature is further enhanced. In addition, the special comb-shaped microheater structure intensifies the heat exchange process and increases the temperature difference.

The response time of the flow sensor is defined as 63.2% of the time it takes for the output value to reach a stable state [32], according to which, the response time of this sensor is obtained to be 35 ms (Figure 6c). Moreover, measurements were tested three times for investigating repeatability and hysteresis of the device, and each measurement was performed with the flow rate increasing first and then decreasing at a step of 100 sccm. The results are displayed in Figure 6d. Correspondingly, the accuracy is a comprehensive reflection of the repeatability and hysteresis of the sensor, which is calculated by the square root of the value obtained by the sum of the squares of the hysteresis and the repeatability error [25]. Thus, the accuracy of the sensor can be calculated as 0.95%. In addition, Figure 6e,f illustrates the long-term stability of the gas flow sensor. The sensitivity of the sensor fluctuated quite slightly (<15 μV/sccm) after several months. Among them, the relative error of multiple measurements is in the range of 2–3%. The results demonstrate that the response of the sensor could remain stable for a long time.

### 3.2. Effects of Chip Angle, Heating Voltage, and Ambient Temperature on Sensor Performance

It is the fundamental characteristic for the thermal flow sensor to realize the measurement of the flow rate according to the difference of heat exchange between the airflow and the heat source. When the heat exchange is not saturated, the larger the heat exchange is, the greater the output change of the thermopile that can be achieved. Differences in the angle at which the direction of the flow intersects the sensor surface may cause discrepancy in heat exchange between the flow and the microheater. Moreover, under the premise that the heat exchange is not saturated, the temperature field generated by the microheater is different, which will also directly affect the heat exchange between the flow and the heat source. It is not difficult to see from the experimental results that output of the sensor changes monotonically with the flow rate, indicating that within the flow rate range of 0–1000 sccm, the airflow heat exchange has not reached the state of saturation. Therefore, the effects of placement angles of the chip and heating voltages on the performance of the flow sensor shall be investigated, which is helpful to determine the optimal conditions of the sensor.

In the measurement, the sensor was placed in the pipeline; the angle between its normal line and the flow direction was set at 0°, 45°, and 90°, respectively. Under these conditions, the response of the sensor to the N_2_ gas with different flow rates was measured. As shown in Figure 7a,b, the sensor is much more sensitive when it is placed perpendicularly to the flow direction. This is because when the flow faces the surface of the microheater, more gas molecules participate in the heat exchange, so that more heat is carried away by the airflow under the same velocity.

Figure 7c,d clearly display that the sensitivity of the sensor is improved as the source voltage is increasing. It is because under a larger heating voltage, the microheater generates more heat, and the surrounding temperature is higher, which induces the heat exchange more intensely. Although the flow rate remains unchanged, more heat is taken away by the airflow. At this time, the difference between the sensor output voltage under a certain flow and the static output voltage is larger. As is illustrated, the difference increases from 69.7 mV (heating voltage of 2 V) to 326.9 mV (heating voltage of 5 V) at a flow rate of 1000 sccm. Consequently, the sensitivity is improved accordingly. However, it should be noted that once the heating voltage is too high, it requires higher energy consumption and may even damage the device structure. Obviously, it is not advisable to load a very high voltage. In order to make it compatible with the working voltage of most microcontrollers, we chose 5 V as the heating voltage, under the premise of the stable and normal operation of the sensor. Accordingly, the power of the microheater was about 46 mW. In addition, we built a test platform with the existing thermostatic chamber (GUANGDONG LIK, LJPTH-225), and in the experiment, temperatures of the thermostat were set at 25 °C, 30 °C, 35 °C, 40 °C, and 45 °C, respectively. As shown in Figure 7e,f, the response of the sensor is influenced by the ambient temperature; specifically, the sensitivity decreases linearly with the increasing ambient temperature. It may be caused by the resistance-temperature characteristics of the aluminum microheater, the different degree of heat exchange between the airflow and the high-temperature air domain in different ambient temperatures, and other factors.

### 3.3. Application of the Sensor in Real-Time Respiration Monitoring

The previous experiments have investigated the detection range, sensitivity, accuracy, and response time of the thermopile-based gas flow sensor; the results indicate that the sensor may be suitable for respiration monitoring. To verify this hypothesis, the sensors were placed in the volunteers’ nostrils to measure respiration airflow. Preliminary tests were conducted on volunteers’ breathing for a period of time, and results are shown in Figure 8. With this system, the breathing states were visualized through the waveforms of the output voltage. Figure 8a displays the waveform image of three breathing cycles. As illustrated in Figure 8b, the respiration monitoring signals from the sensor remained stable after 12 days. Furthermore, the respiration data were collected immediately after a short period of vigorous exercise (Figure 8c). During vigorous exercise, the body respirates rapidly and deeply for obtaining sufficient oxygen, which can be reflected by the larger amplitude and frequency. Meanwhile, the respiration rate decreases gradually; at the same time, the respiration depth becomes shallow and finally returns to the moderate respiration state. Accordingly, the waveform amplitude and frequency both decrease as the respiration changes from rapid to moderate. Figure 8d illustrates the signal changes under different breathing frequencies. The frequency of moderate breathing is about 0.33 Hz; that of deep breathing is relatively slow, about 0.23 Hz; and the frequency of rapid breathing is larger, reaching 0.81 Hz. In addition, Figure 8e,f shows the differences in cycle periods (T) and amplitude variations (D) of the respiration waveforms of the three postures, including lying, sitting and standing. As is shown in the figure, the respiratory cycle for the lying state is the longest, and that for the standing state is the shortest. Furthermore, by randomly selecting 75 respiratory cycles and comparing the periods and amplitude variations, it could be found that there are obvious differences among the postures, as shown in Figure 8g.

## 4. Conclusions

This work presents the design, fabrication, and characterization of a thermopile-based gas flow sensor. The test results show that the gas flow sensor has a normalized detection sensitivity of 6.6 μV/sccm/mW in a flow rate range of 0–1000 sccm, and its response time is only 35 ms. Based on the excellent performance of the sensor, it has been further used for respiration monitoring. The amplitude and frequency of the sensor output signal waveform reflect respiratory at speeds of moderate, rapid, and deep. Moreover, it preliminarily verifies the characteristic differences of human respiration in different postures of lying, sitting and standing. This sensor is able to reflect the respiration states and can distinguish the different activities of a human body, indicating a tremendous potential in personal respiration monitoring.

## Figures and Tables

**Figure 1 micromachines-14-00910-f001:**
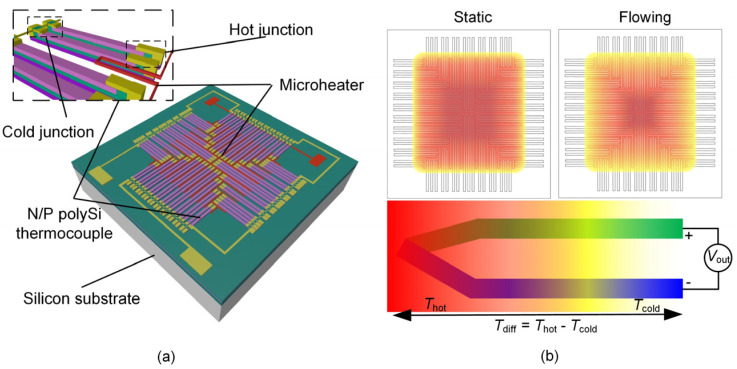
Schematic of a thermopile-based gas flow sensor: (**a**) 3D structure; (**b**) Principle of airflow measurement.

**Figure 2 micromachines-14-00910-f002:**
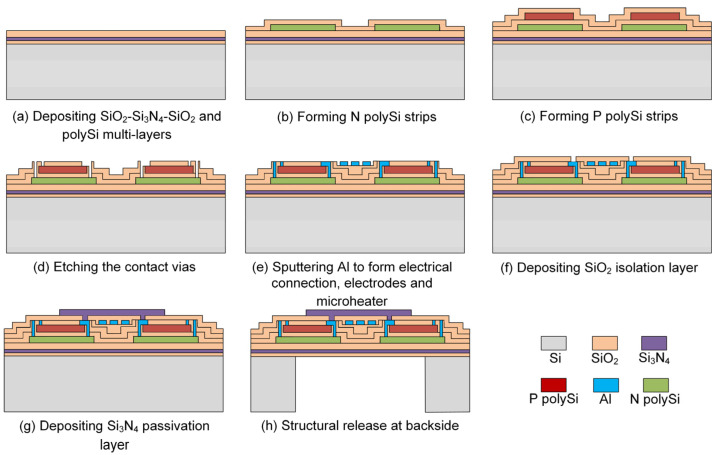
Fabrication process of the thermopile-based gas flow sensor.

**Figure 3 micromachines-14-00910-f003:**
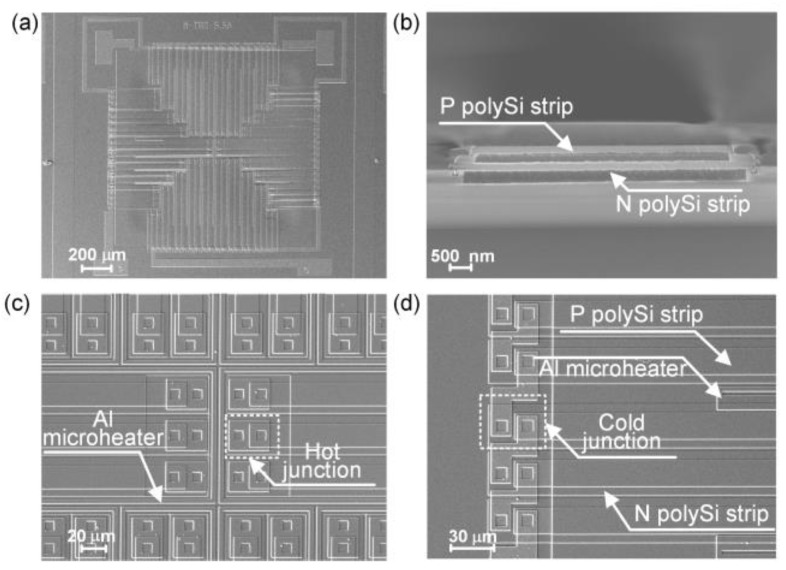
SEM images of the fabricated flow sensor. (**a**) Overall topography of the sensor. (**b**) The cross-section view of a thermocouple. (**c**) SEM image of the hot junctions and microheater. (**d**) SEM image of the cold junctions.

**Figure 4 micromachines-14-00910-f004:**
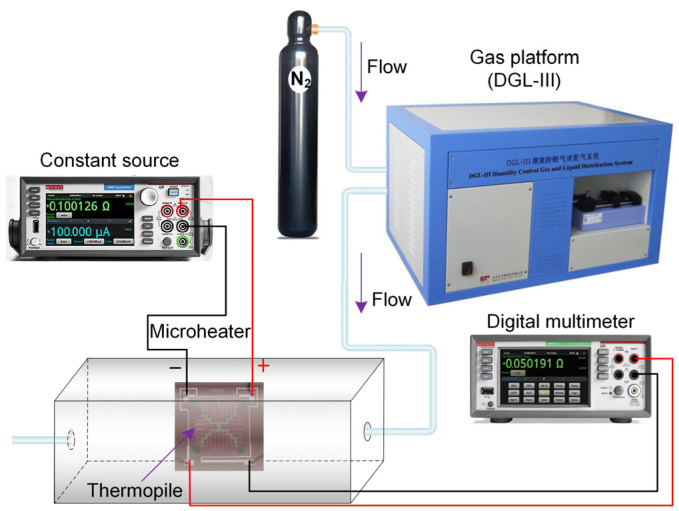
Schematic measurement system for the sensor..

**Figure 5 micromachines-14-00910-f005:**
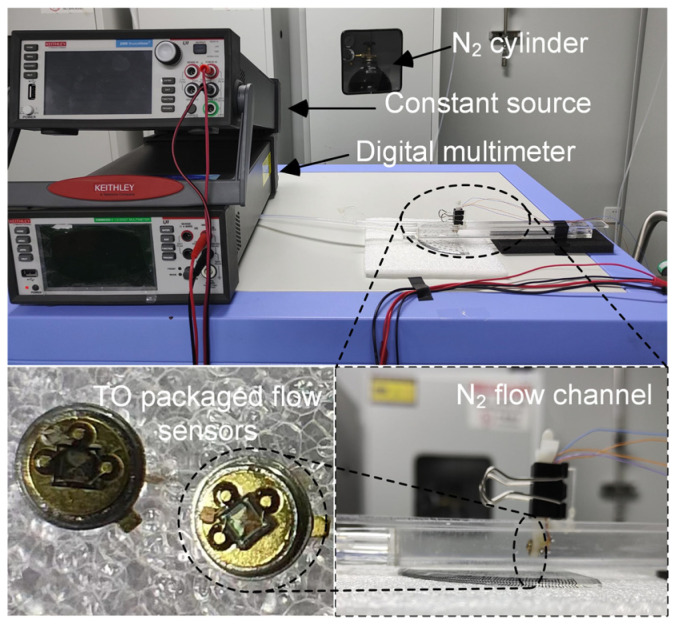
Photographs of the measurement system and the packaged MEMS gas flow sensor.

**Figure 6 micromachines-14-00910-f006:**
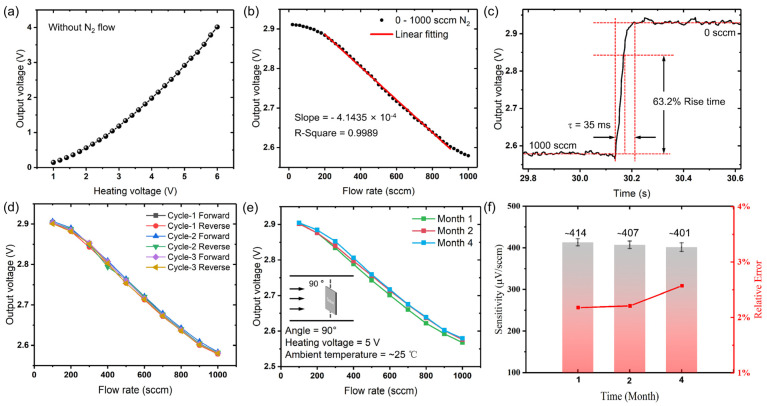
Basic output characteristics of the flow sensor. (**a**) Relationship between the heating voltage and the output voltage. (**b**) Output voltage of the sensor with N_2_ flowing at different rates with a heating voltage of 5 V. (**c**) Response time of the sensor with a heating voltage of 5 V. (**d**) Repeatability and hysteresis of the sensor. (**e**) Long-time stability of the sensors. (**f**) Sensitivities of the sensor in the long-term testing.

**Figure 7 micromachines-14-00910-f007:**
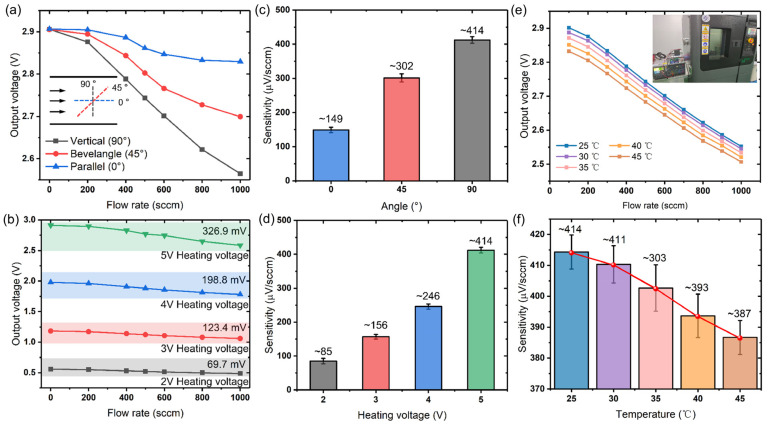
Different factors affecting the performance of the gas flow sensor. (**a**) Response of the sensor at different flow angles. (**b**) Response of the sensor at different heating voltages. (**c**) Sensitivity of the sensor varies with the angle. (**d**) Sensitivity of the sensor varies with the heating voltage. (**e**) Response of the sensor under different ambient temperatures. (**f**) Sensitivity of the sensor varies with the ambient temperature.

**Figure 8 micromachines-14-00910-f008:**
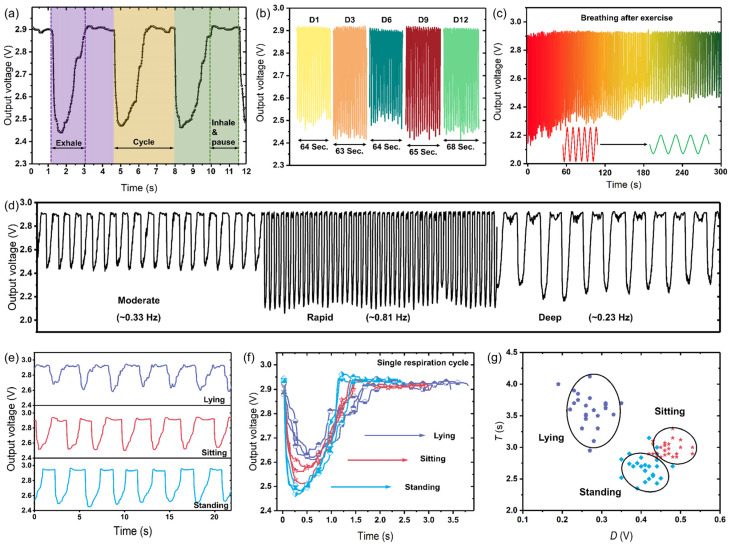
Real-time sensor response to human respiration. (**a**) Voltage curve responding the respiration (three cycles). (**b**) Respiration monitoring results in 12 days. (**c**) Respiration monitoring after vigorous exercise. (**d**) Dynamic responses of the sensor at different respiration states: moderate, rapid, and deep. (**e**) Respiration monitoring of the sensor at three postures: lying, sitting, and standing. (**f**) Waveforms of a single respiration cycle at three postures. (**g**) Visualization map in the respiration at three postures.

**Table 1 micromachines-14-00910-t001:** Structural parameters of the thermopile-based gas flow sensor.

Parameters	Value
Width of the N polySi	30 μm
Thickness of the N polySi	0.4 μm
Width of the P polySi	20 μm
Thickness of the P polySi	0.3 μm
Width of the microheater	5 μm
Thickness of the microheater	0.5 μm
Area of the back cavity	1100 μm × 1100 μm
Size of the sensor	1650 μm × 1650 μm
Number of the N/P polySi thermocouples	78

**Table 2 micromachines-14-00910-t002:** Comparison between the reported sensor and our work.

Structure Type	Suspended Membrane	Size (mm × mm)	Sensitivity (μV/sccm/mW)	Ref.
P + Si/Al	SiN	0.50 × 0.70	0.54	[23]
N/P polySi	Si_3_N_4_-SiO_2_	3.00 × 3.00	0.79	[25]
P + Si/Al	Si-SiO_2_	3.60 × 4.80	0.01	[29]
P polySi/Al	Si	1.10 × 1.50	2.50	[30]
P + Si/Al	SiN-SiO_2_	0.65 × 0.65	0.20	[31]
N/P polySi	SiO_2_-Si_3_N_4_-SiO_2_	1.65 × 1.65	6.62	Our work

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
