# Peer review of "A Thermopile-Based Gas Flow Sensor with High Sensitivity for Noninvasive Respiration Monitoring"

_micromachines, 2023, doi:10.3390/mi14050910_

Round 1

Reviewer 1 Report

This manuscript reports a p+poly/n+poly thermopile-based gas flow sensor and its application in noninvasive respiration monitoring. However, some deficiencies and details should be improved, some comments are as follows.

(1) The author should read the quoted article carefully and avoid errors. For example, the thermopile-based flow sensor in Ref.[23] is p+Si/metal thermopile-based gas flow sensor not polySi/metal flow sensor.

(2) the sensitivity of 414uV/sccm without heater heating-power is difficult to be read. 414uV/sccm/W or 414uV/sccm/mW? the author must point out clearly. 

(3) I have some doubts about Eq.1-3. When airflow flows through the sensor, can the output voltage of the sensor be expressed as Δ U=( α P- α N) Δ T?

(4) in line 117, confirm that Si3N4 is not a high thermal resistance, but a high thermal conductivity?

(5) in line 176-177, the performance comparison results with the previously reported thermopile-based sensors should be listed in a table, which is convenient for readers to understand. in addition, the thermopile-based gas flow sensor in ref. [23] should also be added to the comparison.

(6) in line 193-196, In general, accuracy should not only consider hysteresis and repeatability, but also take account of nonlinearity.  Therefore, is the accuracy evaluation method reasonable in this manuscript?

(7) In Fig.7a, when the airflow faces the surface of the microheater, does the author consider the effect of the airflow-velocity-induced the turbulence of the insulation film on the detection accuracy? Especially for high flow rate gas, which will make film rupture.

(8) As shown in Fig.7e, the output voltage changes with the ambient temperature. How to ensure the detection accuracy in the subsequent application? using CTD compensated circuit or other?

Author Response

Thank you for your helpful comments on our article. We have carefully responded to the reviewers' comments one by one, please see the attachment.

Reviewer 2 Report

The article deals with an interesting approach to measure the gas flow in breathing gas analysis with a thermoelectric sensor built in microsystem technology or Si technology.  Some questions should be answered before the article can be published (minor revision):

1. page3, line 101: What is the Seebeck coefficient of the used thermoelectric materials p-polySi and n-polySi?

2. page6, figure 6: An explanation should be included why the measured (thermoelectric) voltage decreases with increasing gas flow.
   - To what temperature gradient across the sample does the measurable thermoelectric voltage or output voltage correspond?  
- What temperature or temperature gradient does 5V heating voltage correspond to, and what is the change in temperature gradient due to the varying gas flow rates?
The authors should comment on these points.

3. page6, Fig. 6(b): The authors should discuss the non-linear behavior of the output voltage at low flow rates and high flow rates. In Fig. 6(c), the meaning of angle = 90° should be explained.

4. page 6, line 198: It is stated, that the sensitivity fluctuation is smaller than 15µV/sccm. How large is the resulting error on the sccm determination?

5. chapter 3.2: Did you investigate the temperature distribution over the sensor in dependence on the flow rates and the heating voltage applied to the microheater? Did you find any influences of changes in the temperature distribution on the measured signal?

6. Fig. 7: Please check the labeling of Fig. 7 (a...f).

7. Fig. 7(a),(b): How does the absolute temperature or the temperature gradient in the sensor change depending on the angle of gas flow and does this also influence the temperature distribution of the sensor?

7. page 7: Why changes the sensitivity in dependence on the ambient temperature? Please comment on this.

8. Chapter 4: The conclusion should be a bit more detailed, especially regarding the possible application.

Author Response

(The authors gave the same response as above.)

Round 2

Reviewer 1 Report

(1)Since the sensor measured-range unit is sccm, it is recommended to correct the sensitivity unit to mV/sccm/mW for the convenience of readers.

(2)In line 121, the high thermal conductivity should be corrected to low thermal conductivity in the manuscript.

Author Response

Thank you again for your useful comments of our article. We have carefully responded to the reviewers' comments one by one, please see the attachment.
